# AbsGS: Recovering Fine Details for 3D Gaussian Splatting

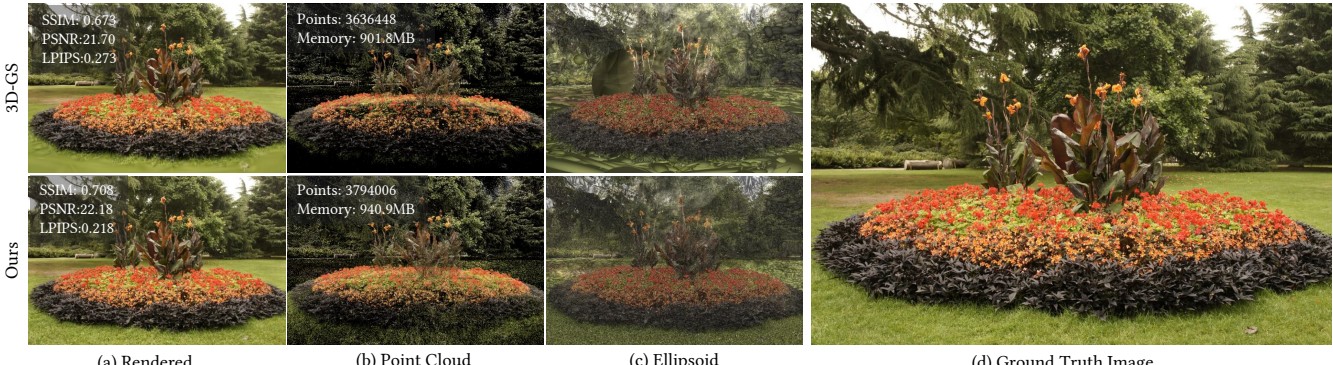

(a) Rendered   (b) Point Cloud   (c) Ellipsoid   (d) Ground Truth Image

**Figure 1: We reveal that the original adaptive density control strategy in 3D Gaussian Splatting (3D-GS) has the flaw of gradient collision which results in degradation, and propose homodirectional gradient as the guidance for densification. (a) Our method recovers fine details and achieves higher quality novel view synthesis results. SSIM, PSNR, LPIPS are inset. (b) Our proposed method yields more reasonable distribution of Gaussion points with comparable number of Gaussians and memory consumption with 3D-GS. (c) By adopting our method, the large Gaussians in over-reconstructed regions that lead to blur are eliminated.**

## ABSTRACT

3D Gaussian Splatting (3D-GS) technique couples 3D Gaussian primitives with differentiable rasterization to achieve high-quality novel view synthesis results while providing advanced real-time rendering performance. However, due to the flaw of its adaptive density control strategy in 3D-GS, it frequently suffers from over-reconstruction issue in intricate scenes containing high-frequency details, leading to blurry rendered images. The underlying reason for the flaw has still been under-explored. In this work, we present a comprehensive analysis of the cause of aforementioned artifacts, namely gradient collision, which prevents large Gaussians in over-reconstructed regions from splitting. To address this issue, We propose the novel homodirectional view-space positional gradient as the criterion for densification. Our strategy efficiently identifies large Gaussians in over-reconstructed regions, and recovers fine details by splitting. We evaluate our proposed method on various challenging datasets. The experimental results indicate that our approach achieves the best rendering quality with reduced or similar memory consumption. Our method is easy to implement and can be incorporated into a wide variety of most recent Gaussian Splatting-based methods. We will open source our codes upon formal publication.

## CCS CONCEPTS

• **Computing methodologies** → **Reconstruction**; **Rendering**.

## KEYWORDS

Novel View Synthesis, 3D Gaussian Splatting, Point-based Radiance Field, 3D reconstruction

## 1 INTRODUCTION

High quality novel view synthesis from multiple unordered images is a long-standing problem for 3D vision researchers. Recent advances on neural rendering have revolutionized this task by learning a neural implicit representation instead of explicit point clouds or meshes. One of the most effective approach with in this paradigm has been reconstructing a set of 3D Gaussian primitives of the scene[17]. Coupled with splat-based rasterization, 3D Gaussian Splatting (3D-GS)[17] produces compelling real-time rendering results with unprecedented fidelity. The remarkable performance of 3D-GS is closely tied to the adaptive density control strategy. Initialized solely from a set of sparse point clouds derived from Structure from Motion (SfM), 3D-GS gradually populate empty areas by split/cloning existing Gaussians, ultimately covering whole scenes with compact and precise representation. There have been many interests on extending 3D-GS to other applications, e.g., dynamic modeling[9, 29, 37, 42], single-view or text-to-view generation[30, 39, 43, 44], mesh extraction[12, 15, 32], SLAM[8, 14, 40] and so on.

However, applying 3D Gaussian Splatting to complex scenes encounters the issue of over-reconstruction, where regions containing high frequency details are covered by only a small number of large Gaussians. Consequently, the rendering results become blurry and

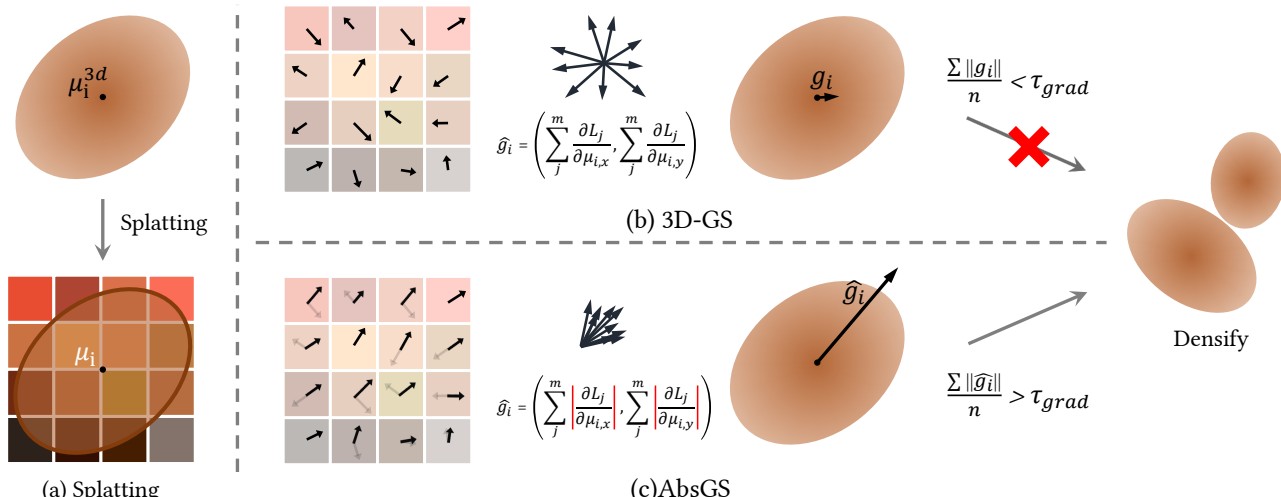

Figure 2: Overview of our method. (a) The splat-based rendering technique project Gaussian $G_i$ with mean position $\mu_i^{3d}$ to 2D coordinate $\mu_i$ in pixel-space. The number of covered pixels by Gaussian $G_i$ is $m$. (b) By backpropagating, the view-space gradient $g_i$ of Gaussian $G_i$ under viewpoint $k$ is caculated as the sum of all view-space gradients of pixels that are covered by $G_i$. Since the gradients $\frac{\partial L_j}{\partial \mu_i}$ have different directions, the overall sum $g_i$ will have a small scale, which do not satisfy the gradient threshold for densification. (c) Motivated by above analysis, we redesign densitifaction scheme by taking the absolute value of each component $|\frac{\partial L_j}{\partial \mu_{i,x}}|$ and $|\frac{\partial L_j}{\partial \mu_{i,y}}|$ before summing. This enables to identify large Gaussians in over-reconstructed regions for split.

cannot accurately reflect the appearance and geometry of the scene as validated in Fig. 1. The cause of over-reconstruction is that the adaptive density control strategy cannot effectively identify large Gaussians in over-reconstructed areas and split them to represent details. The deficiency of the strategy has not been well explored.

In this paper, we find that the deficiency of the original strategy lies in its failure to consider the negative impact of pixel-wise sub-gradient directions on the identification of large Gaussians in over-reconstructed areas. Specifically, original 3D-GS proposes to use the view-space positional gradient to determine whether if a large Gaussian requires split. We observe that for each Gaussian primitive the pixel-wise sub-gradients of the view-space positional gradient may have different directions. Therefore, the sub-gradients cancel each other out during the process of summation, namely *gradient collision*. Especially for large Gaussians covering many pixels, maintaining consistent gradient directions for each pixel becomes exceptionally challenging which results in a small-scale view-space positional gradient. Consequently, the magnitude of view-space positional gradient fails to surpass the densification threshold, thereby hindering the split of over-reconstructed Gaussians.

Based on the above analysis, we propose homodirectional view-space positional gradient as criteria for densification. Homodirectional view-space positional gradient is designed as the sum of the absolute values of pixel-wise sub-gradients covered by a Gaussian primitive, based on the rationale that the representation quality is solely dependent on the magnitude of the gradient, irrespective of its direction. The absolute operation can mitigate the influence of

gradient direction while retaining the influence of gradient magnitude. The homodirectional view-space positional gradient therefore avoids the gradient collision and facilitates the split of large Gaussians in over-reconstructed regions that were unrecognized by original strategy. An overview of our method (dubbed as AbsGS) is shown in Fig. 2.

We evaluate AbsGS on previously published real-world datasets. The experiment results show that our method consistently yields high quality novel view synthesis and exhibits better results on PSNR, SSIM, and LPIPS. At the same time, our method keeps similar or less memory consumption compared with 3D-GS. From the visualization of Gaussian ellipsoids, we observe that our method eliminates over-reconstruction areas and recovers fine details while 3D-GS fails and leads to blur, as illustrated in Fig. 1. In summary, our contributions are as follows:

- First, we analyze the deficiency of the original strategy that results in over-reconstruction is caused by gradient collision.
- Second, a straightforward yet effective strategy is proposed to utilize homodirectional view-space positional gradient as guidance for densification throughout training.
- Third, our proposed method can effectively eliminate large Gaussians in over-reconstructed regions, and achieves better novel view synthesis quality with similar or less memory consumption.

## 2 RELATED WORKS

*Neural Implicit 3D Representation.* In contrast to widely adopted classic explicit 3D representaiton, e.g., point cloud, voxels and

mesh, more recent learning-based neural implicit representations do not require complex regularization, attract more attention and achieve more accurate rendering results. As a revolutionary pioneer, Neural Radiance Fields (NeRF)[20] couples differentiable ray-marching with continuous radiance field to enable end-to-end optimization from images. NeRF has been broadly receiving massive interest towards more photo-realistic novel view synthesis[1, 2, 4] and its follow-up methods have been providing impressive results to other applications, e.g, dynamic modeling[11, 24, 25], surface reconstruction[35, 36, 38, 41], 3D asset generation[16, 26, 34]. However, with expensive volumetric ray-marching which densely sample point locations along the camera rays, NeRF is inefficient for training at the beginning of design. Though notable NeRF-variants [5, 10, 21, 27] have been proposed to alleviate the training/inference computation burden by introducing spatial datastructures to store neural features instead of large MLPs, the plenty of sampling and queries can not be avoided due to the inherit requirement of volumetric rendering. Instead, point based representations with learnable attributes support more efficient forward rasterization for real-time rendering. More recently, 3D Gaussian Splatting (3D-GS)[17], revisit fast point-based rendering engine with learnable Gaussian primitives. Starting from initial sparse point clouds from Structure from Motion (SfM)[23, 28, 33], the optimization procedure moves the Gaussians to correct positions, creates new Gaussians to cover empty space, removes invalid Gaussians and finally produces a set of Gasussions to precisely represent underlying scenes. As an unstructured and discrete representation that supports forward rasterzation, it fundamentally avoid the shortcomings of expensive sampling and queries and provide real-time rendering performance, along with high quality for novel-view synthesis. Nowadays there have been many subsequent extensions of 3D Gaussian splatting, e.g, surface reconstruction[6, 12, 15], generation[7, 22, 31, 39] and dynamic modeling[19, 37].

## 3 METHOD

In this section, we first review the basic background of 3D-GS in Section 3.1; then, we describe the gradient collision phenomenon that prevent large Gaussions in over-reconstructed regions from splitting in Section 3.2; finally, we propose the homodirectional gradient as guidance for splitting and present the details in Section 3.3.

### 3.1 Preliminary

3D Gaussian Splatting (3D-GS)[17] proposes to represent scenes by a set of learnable 3D Gaussians $G_0, G_1, ..., G_N$. Each 3D Gaussian primitive $G_i$ is explicitly parameterized via center position $\mu_i^{3d}$ and full 3D covariance matrix $\Sigma_i^{3d}$:

$$G_i(x) = e^{-\frac{1}{2}(x-\mu_i^{3d})^T (\Sigma_i^{3d})^{-1}(x-\mu_i^{3d})} \quad (1)$$

3D Gaussian primitive also has two additional learnable attributes: opacity $o_i$ and spherical harmonics coefficients $SH_i$ to model view dependent color. To render an image, the 3D Gaussian primitive $G_i$ is first transformed into the camera coordinate and projected onto image plane, resulting the 2D Gaussian $G_i^{2d}$ with center position $\mu_i$ and 2D covariance matrix $\Sigma_i^{2d}$:

$$G_i^{2d}(x) = e^{-\frac{1}{2}(x-\mu_i)^T (\Sigma_i^{2d})^{-1}(x-\mu_i)} \quad (2)$$

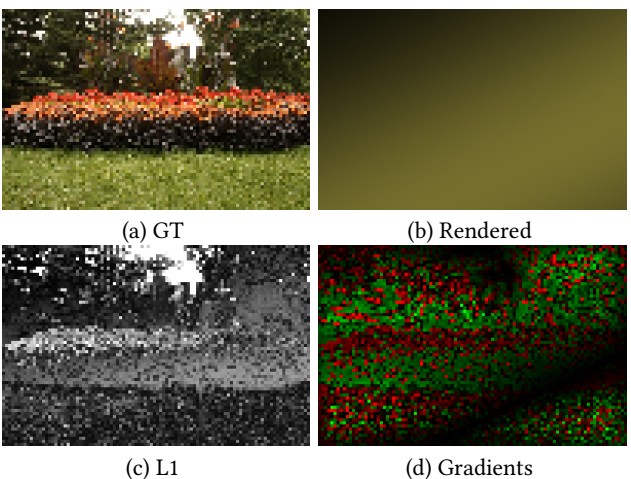

(a) GT      (b) Rendered

(c) L1      (d) Gradients

**Figure 3: We analyze gradient collision for view-space positional gradient, by optimizing single Gaussian to fit a image. We show the x-axis direction of pixel-wise gradient in (d), where red represents positive and green represents negative.**

then differentiable alpha blending is employed to integrate colors from front-to-back:

$$c(x) = \sum_i^N c_i \alpha_i \prod_{j=1}^{i-1} (1 - \alpha_j) \quad (3)$$

$$\alpha_i = \sigma(o_i) \times G_i^{2d}(x) \quad (4)$$

where $\sigma(\cdot)$ is the sigmoid function and $\mathcal{N}$ is the number of Gaussians that participate in alpha blending. 3D-GS initializes 3D Gaussians with the free sparse point clouds produced from SfM[28], and then applies adaptive density control to populate empty areas. There are two forms of densification: split and clone. Split operation is designed to split large Gaussians that represent small-scale areas in two, which corresponds to *over-reconstruction*. Clone operation aims at clone more Gaussians to sufficiently cover *under-reconstruction* region For each Gaussian $G_i$, 3D-GS uses the average magnitude of view-space positional gradients to determine whether to apply densification. Specifically, for Gaussian $G_i$ which have the pixel-space projection point $\mu_i^k = (\mu_{i,x}^k, \mu_{i,y}^k)$ under viewpoint $k$ and corresponding loss $L^k$, the average view-space positional gradient $\nabla_{\mu_i} L$ is calculated every 100 training iterations as follows:

$$\nabla_{\mu_i} L = \frac{\sum_{k=1}^M ||\frac{\partial L^k}{\partial \mu_i^k}||}{M} = \frac{\sum_{k=1}^M \sqrt{(\frac{\partial L^k}{\partial \mu_{i,x}^k})^2 + (\frac{\partial L^k}{\partial \mu_{i,y}^k})^2}}{M} \quad (5)$$

where $M$ is the total number of viewpoints that Gaussian $G_i$ participates in calculation during 100 iterations. Split for $G_i$ is performed when it satisfies:

$$\nabla_{\mu_i} L > \tau_p \text{ and } \Sigma_i^{3d} > \tau_S \quad (6)$$

where $\tau_p$ is the the gradient threshold (default 0.0002) and $\tau_S$ is the scale threshold (default 0.01).

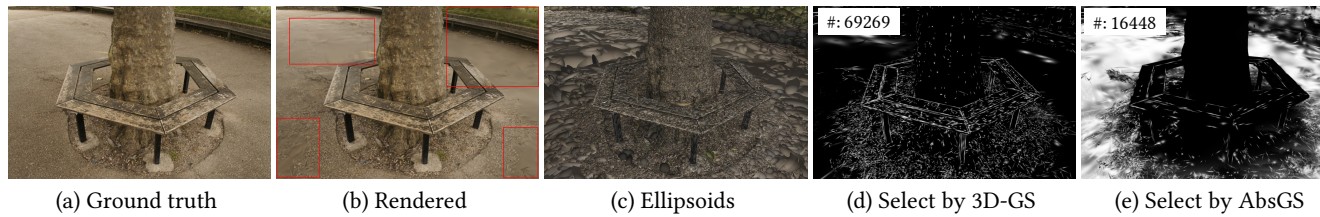

|  (a) Ground truth | (b) Rendered | (c) Ellipsoids | (d) Select by 3D-GS | (e) Select by AbsGS |

**Figure 4: An example to demonstrate the difference between densification strategy of 3D-GS and AbsGS. From (b) and (c), we observe that large-scale Gaussians are used to represent cement ground, which contains fine details and indeed should be represented by many small-scale Gaussians. In (d) and (e), we show the Gaussians that satisfy densification criteria of 3D-GS and ours respectively, where identified Gaussians' colors are set to white. When using $g_i$ as 3D-GS, the large Gaussians that represent cement ground are not identified while our selection strategy based on $\hat{g}_i$ can find those Gaussians.**

## 3.2 Gradient Collision

3D-GS relies on gradient descent to optimize the scene, so the magnitude of the gradient can reflect the quality of representation. However, the computation of $\nabla_{\mu_i} L$ includes gradient directions that are irrelevant to the representation state, weakening the effectiveness of the gradient magnitude. In this section, we analyze the negative impact of gradient directions. Specifically, there exists *gradient collision* in the calculation of $\frac{\partial L^k}{\partial \mu_{i,x}^k}$ and $\frac{\partial L^k}{\partial \mu_{i,y}^k}$, thus affect the role of $\nabla_{\mu_i} L$. To simplify the notation, we discard view $k$ in following notations, and we use $g_i$, $g_{i,x}$ and $g_{i,y}$ to refer $\frac{\partial L}{\partial \mu_i}$, $\frac{\partial L}{\partial \mu_{i,x}}$ and $\frac{\partial L}{\partial \mu_{i,y}}$ respectively.

Take the x-axis gradient $g_{i,x}$ as an example. We further decompose this gradient into the sum of multiple sub-gradients contributed by each pixel:

$$g_{i,x} = \frac{\partial L}{\partial \mu_{i,x}} = \sum_{j=1}^{m} \frac{\partial L_j}{\partial \mu_{i,x}} \tag{7}$$

where $m$ is the number of pixels covered by $G_i$, $L_j$ is the loss computed by $j$-th pixel. Our key observation is that the per-pixel gradients $\frac{\partial L_j}{\partial \mu_{i,x}}$ may have different directions. The proof is as follows.

*Proof.* We simplify the proving goal to demonstrating that the signs of gradients may differ, which is equivalent to proving that the gradient directions are different. Per-pixel gradient $\frac{\partial L_j}{\partial \mu_{i,x}}$ can be calculated as:

$$\frac{\partial L_j}{\partial \mu_{i,x}} = \sum_{l=1}^{3} \frac{\partial L_j}{\partial c_l^j} \times \frac{\partial c_l^j}{\partial \alpha_i} \times \frac{\partial \alpha_i}{\partial \mu_{i,x}} \tag{8}$$

The sign of $\frac{\partial L_j}{\partial \mu_{i,x}}$ is determined by the multiplication of signs of three individual terms.

For the first term, since $L_j$ is $\mathcal{L}_1$ loss, the sign of first term $\frac{\partial L_j}{\partial c_l^j}$ depends on the comparison result of rendered and real RGB values.

For the second term $\frac{\partial c_l^j}{\partial \alpha_i}$, according to Equ.3, it can be further calculated as follows:

$$\frac{\partial c_l^j}{\partial \alpha_i} = \prod_{l=1}^{i-1} (1 - \alpha_l) c_i + \sum_{p=i+1}^{N} c_p \frac{\partial w_p}{\partial \alpha_i}, \tag{9}$$

$$\frac{\partial w_p}{\partial \alpha_i} = -\alpha_p \prod_{l=1, l \neq i}^{p-1} (1 - \alpha_l), \tag{10}$$

Thus, the two additions to Equ.9 have opposite signs, and the sign of $\frac{\partial c_l^j}{\partial \alpha_i}$ is uncertain. Here $c_i$ and $c_p$ are Gaussian colors, $c_l^j$ is the color of the pixel.

For the third term $\frac{\partial \alpha_i}{\partial \mu_{i,x}}$, according to Equ.4, it is calculated as:

$$\frac{\partial \alpha_i}{\partial \mu_{i,x}} = \sigma(o_i) \times \frac{\partial G_i^{2d}(\mu_{i,x})}{\partial x} = \sigma(o_i) \times \frac{(\mu_{i,x} - \hat{p}_j) G_i^{2d}(\mu_{i,x})}{\sigma_1^2} \tag{11}$$

where $\hat{p}_j$ is the coordinate of $j$-th pixel. Therefore, the sign of the third term $\frac{\partial \alpha_i}{\partial \mu_{i,x}}$ is determined by the relative x-axis coordinate difference $(\mu_{i,x} - \hat{p}_j)$.

Overall, the sign of first term is determined by to the rendered pixel value, the sign of second term is related all the Gaussians's that participate in calculation, and the sign of third term is related to the projection point, so these three terms do not always have the same sign, and the per-pixel gradient $\frac{\partial L_j}{\partial \mu_{i,x}}$ may have different directions for different pixels.

We design a simple experiment to verify the above analysis, as illustrated in 3. An image is randomly selected from the *flowers* scene from Mip-NeRF360[3] and is reduced to a resolution of 100×65 in Fig. 3 (a). Then we optimize only one Gaussian to fit this image, and the final rendering result is shown in Fig. 3 (b). The overall L1 loss is large and this Gaussian leads to a typical over-reconstruction issue, in Fig. 3 (c). we show the x-axis gradient direction $\frac{\partial L_j}{\partial \mu_{i,x}}$ in Fig. 3 (d), where the pixel color means the x-axis direction of the gradient contributed by this pixel, and red is for positive x-axis direction and green for negative x-axis direction. The different directions of per-pixel gradient $\frac{\partial L_j}{\partial \mu_{i,x}}$ results the sum $\nabla_{\mu_i} L$ may have a small-scale magnitude . Especially for large Gaussians that covering many pixels, maintaining consistent gradient directions for each pixel becomes exceptionally challenging. Consequently,

Table 1: Quantitative results on Mip-NeRF 360[1], Tanks & Temples[18], and Deep Blending[13]. All scores of the rest of the baselines are directly sourced from the original 3D-GS paper[17] to make a fair comparison. INGP-Base and INGP-Big are Instant-NGP versions with default settings and increased network size respectively. The 1st, 2nd, and 3rd-best performances are indicated by red, orange, and yellow highlights respectively.

| Datasets | Mip-NeRF360 | | | | Tanks & Temples | | | | Deep Blending | | | |
|---|---|---|---|---|---|---|---|---|---|---|---|---|
| Methods | SSIM | PSNR | LPIPS | Mem | SSIM | PSNR | LPIPS | Mem | SSIM | PSNR | LPIPS | Mem |
| Plenoxels | 0.626 | 23.08 | 0.463 | 2.1GB | 0.719 | 21.08 | 0.379 | 2.3GB | 0.795 | 23.06 | 0.510 | 2.7GB |
| INGP-Base | 0.671 | 25.30 | 0.371 | 13MB | 0.723 | 21.72 | 0.330 | 13MB | 0.797 | 23.60 | 0.423 | 13MB |
| INGP-Big | 0.699 | 25.59 | 0.331 | 48MB | 0.745 | 21.92 | 0.305 | 48MB | 0.817 | 24.96 | 0.390 | 48MB |
| Mip-NeRF360 | 0.792 | 27.69 | 0.237 | 8.6MB | 0.759 | 22.22 | 0.257 | 8.6MB | 0.901 | 29.40 | 0.245 | 8.6MB |
| 3D-GS | 0.815 | 27.21 | 0.214 | 734MB | 0.841 | 23.14 | 0.183 | 411MB | 0.903 | 29.41 | 0.243 | 676MB |
| 3D-GS* | 0.809 | 27.36 | 0.220 | 760MB | 0.842 | 23.64 | 0.179 | 374MB | 0.897 | 29.57 | 0.240 | 624MB |
| AbsGS-0008 | 0.815 | 27.41 | 0.211 | 450MB | 0.844 | 23.54 | 0.1831 | 202MB | 0.903 | 29.69 | 0.241 | 380MB |
| AbsGS-0004 | 0.820 | 27.49 | 0.191 | 728MB | 0.853 | 23.73 | 0.162 | 304MB | 0.902 | 29.67 | 0.236 | 444MB |

the gradient magnitude fails to surpass the densification threshold $\tau_p$, thereby hindering the split of over-reconstructed Gaussians.

## 3.3 Homodirectional Gradient

Based on the above analysis, we design AbsGS, which can accurately reflects the representation state and identify Gaussians in over-reconstructed regions. The overall review of our method is shown in Fig. 2. AbsGS aims to eliminate gradient collision by erasing the influence of gradient direction while retaining only the influence of gradient magnitude. Specifically, AbsGS computes the homodirectional view-space positional gradient $\hat{g}_i$ by taking the absolute value of each component before summing:

$$\hat{g}_i = (\hat{g}_{i,x}, \hat{g}_{i,y}) \tag{12}$$

$$\hat{g}_{i,x} = \sum_{j=1}^{m} |\frac{\partial L_j}{\partial \mu_{i,x}}|, \quad \hat{g}_{i,y} = \sum_{j=1}^{m} |\frac{\partial L_j}{\partial \mu_{i,y}}| \tag{13}$$

The absolute operation constrains the gradient directions of all pixels to be in the same direction along the x and y axes, thereby avoiding gradient collision. It uses the magnitudes of the gradient components along the x and y axes to jointly express the state of representation, and finally combines these two components through the $L2$ norm. This value directly reflects the expression state of all the pixels covered by the Gaussian, thus accurately identifying Gaussians with subpar expression, such as over-reconstructed Gaussians. Note that the $\hat{g}_i$ is not used in backpropagation of computation graph, it is an extra variable that only related to densification.

Fig. 4 presents a real example to compare the selected Gaussians that need densification between 3D-GS and ours. We first optimize the scene *treehill* of Mip-NeRF360 for 7000 iterations using 3D-GS. then we select over-reconstructed Gaussians by original $g_i$ and ours homodirectional $\hat{g}_i$. The selected Gaussians are highlight with white, as shown in Fig. 4 (d) and (e). It shows that 3D-GS selects 69,269 Gaussians for densification but it missed most of the large Gaussians in over-reconstructed areas. In contrast, our proposed AbsGS only select 16,488 Gaussians while covering the majority of over-reconstructed areas. This experiment effectively demonstrates that our method is better suited as the criteria for densification.

## 4 EXPERIMENTS

### 4.1 Setup

*Datasets.* Following 3D-GS[17], we select scenes with highly diverse capture styles, spanning from enclosed indoor environments to expansive outdoor settings without clear boundaries. Specifically, we use all 9 unbounded indoor and outdoor scenes presented in Mip-NeRF360[3], two scenes from [18] and two scenes provided by [13], totaling 13 distinct environments.

*Baselines.* Across all dataset, we benchmark our proposed method against multi-level hash grid based Instant-NGP[21], the current state-of-the-art NeRF-based method Mip-NeRF360, and widely-used 3D GS[17]. To ensure a fair comparison, all numerical data of these methods presented in tables are directly sourced from the original publication[17] unless otherwise specified. In addition, we observe that the scale of some scene details is smaller than the default radius threshold $\tau_S$ that used to control split operation, and thus 3D-GS is hindered in sufficient split for these regions. To further improve enhance the ability of 3D-GS. we reduced the $\tau_S$ from default 0.01 to 0.001 and retrain 3D-GS, which is denoted as 3D-GS* in following experiments. Our method AbsGS is also trained with $\tau_S$ set as 0.001. Since $\tau_p$ is a hyper-parameter that impact the behavior of AbsGS, we present the results when $\tau_p$ is set to different values. Specifically, AbsGS-0004 and AbsGS-0008 represent the results when $\tau_p$ is set to 0.0004 and 0.0008 respectively.

*Implementation Details.* Our experiments is measured on a single NVIDIA V100 GPU with 32GB memory. Following common practice, we stop the Gaussian densification after 15k iterations and stop training at 30k iterations. We report the novel view synthesis metrics PSNR, SSIM, and LPIPS, and we also show the number of Gaussians and the memory used to store the parameters of optimized Gaussians to demonstrate the trade-off between efficiency and memory. To highlight the effectiveness of AbsGS in addressing over-reconstruction, homodirectional gradients are only used as guidance for split operation while clone operation follows the original strategy of 3D-GS directly. We select a larger gradient threshold $\tau_p$ for split operation since AbsGS will increase the magnitude of $\nabla_{\mu_i} L$ than that in 3D-GS.

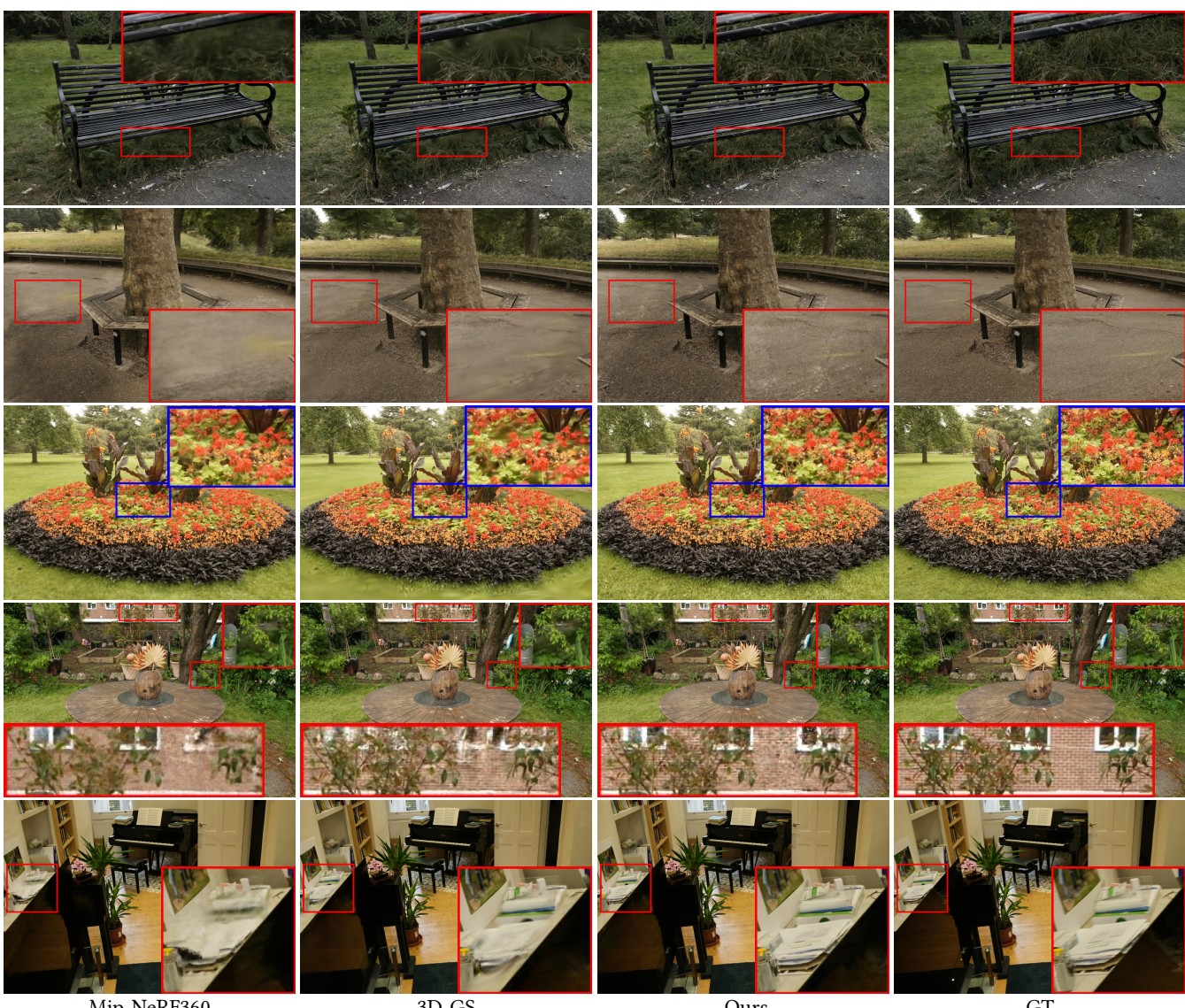

Mip-NeRF360        3D-GS        Ours        GT

**Figure 5: Qualitative comparisons of different methods on scenes from Mip-NeRF360[3] and Tanks&Temples[18] datasets. The rendering result of 3D Gaussian Splatting is blurry at regions containing high-frequency details. Our AbsGS yields significantly better rendering quality with sharper details.**

## 4.2 Performance Evaluation

*Quantitative Results.* We report the quantitative results in Table. 1. Both AbsGS-0004 and AbsGS-0008 outperform other baselines in most cases. It's noteworthy that our method consistently yields better result in terms of SSIM and LPIPS metrics, which capture more reliable human perception differences in images than PSNR metric. Additionally, it can be observed from the memory consumption that the effectiveness of AbsGS does not stem from an increased number of Gaussians. Compared to original 3D-GS, AbsGS-0008 only maintains roughly half the memory consumption and AbsGS-0004 also consistently maintains lower memory, while the quantitative result of them remain highly competitive.

*Qualitative Results.* First, we show novel view synthesis results in Fig. 5. The results indicate that our method significantly reduces the rendering blurring phenomenon and improves rendering quality across all scenes, such as weeds under the bench and uneven concrete ground. Second, to demonstrate the effectiveness of our method in eliminating large Gaussians in over-reconstructed regions, we select *stump* scene and visualize the point clouds and ellipsoids along with the number of Gaussians in Fig. 7. We can observe that 3D-GS exhibits over-reconstruction in areas with similar colors but rich textures, such as the grassy area, where the point cloud is very sparse and the ellipsoids are excessively large. In contrast, our method effectively utilizes smaller Gaussians for

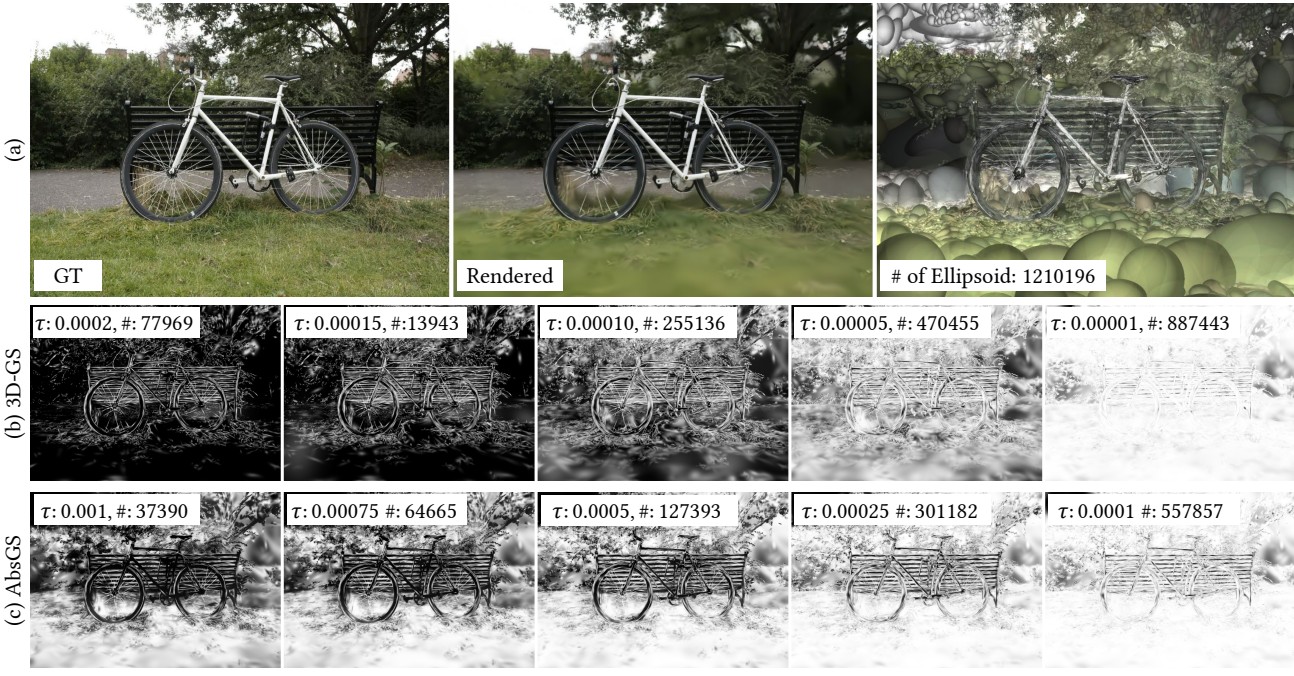

**Figure 6: Comparison of identified Gaussians under different gradient thresholds. (a) The result of training the *bicycle* scene for 3000 iterations using 3D-GS with a gradient threshold of 0.0002, showing significant over-reconstruction. (b) and (c) respectively show the selection results of 3D-GS and AbsGS with different gradient thresholds at this stage. White represents Gaussians that are selected for densification while black represents those that do not. The threshold and the number of selected Gaussians are both annotated.**

representing those areas. It is worth mentioning that the number of Gaussians of AbsGS is less than that of 3D-GS, with Abs-0008 even being less than half of 3D-GS. This indicates that our method does not rely on more Gaussians to solve the problem of over-reconstruction. Additionally, the scale threshold for Abs-0008* is set to 0.01 as same as 3D-GS, to demonstrate that the effectiveness of our method does not depend on adjusting the hyperparameter. We provide more qualitative comparisons in the supplementary material.

*Ablation Studies.* In this section, we conduct ablation experiments to study the impact of scale threshold $\tau_S$ and gradient threshold $\tau_p$ on our method. We present quantitative results in in Tab.2. The experimental results demonstrate that lowering any of $\tau_p$ and $\tau_S$ improves rendering quality at the cost of memory. Additionally, to qualitatively illustrate the impact of $\tau_S$ on rendering quality, we show rendering result in Fig.8. The visualization demonstrates that a smaller $\tau_S$ helps to mitigate large Gaussians in over-reconstructed regions. The reason is that the scale of some scene details is smaller than $\tau_S$; consequently, the Gaussions on those regions that have larger radius than $\tau_S$ will not split and thus lead to blurry rendering.

## 4.3 Analysis

*Why not solve the over-reconstruction problem by simply lowering the threshold $\tau_p$?* Since over-reconstruction issue is caused by

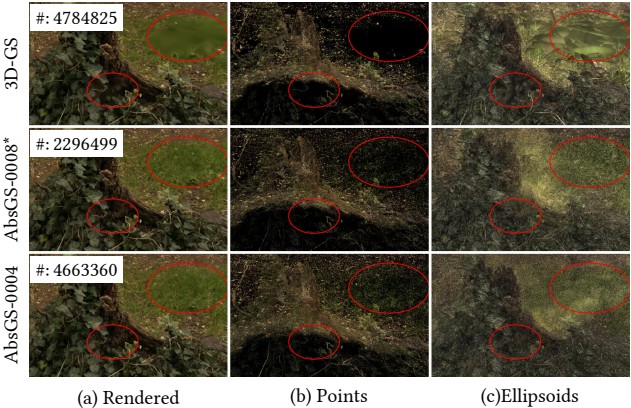

(a) Rendered      (b) Points      (c)Ellipsoids

**Figure 7: The visualization of rendered images, point clouds, and ellipsoids. The scale threshold for both AbsGS-0008* and 3D-GS is 0.01, while the scale threshold for AbsGS is 0.001.**

gradient collision that make it difficult for the gradient of large Gaussions to exceed threshold $\tau_p$, a straightforward solution is lowering threshold $\tau_p$ to identify more large Gaussians in over-reconstructed regions. However, we have observed that this solution doesn't perform well in practice and leads to significant memory consumption. Fig. 9 (a) shows the LPIPS and memory consumption of 3D-GS

**Table 2: Quantitative studies for different scale threshold $\tau_S$ and gradient threshold $\tau_p$. The 1st and 2nd performances are indicated by red and orange highlights respectively.**

| Datasets | | Mip-NeRF360 | | | | Tanks & Temples | | | | Deep Blending | | | |
|---|---|---|---|---|---|---|---|---|---|---|---|---|---|
| Methods | | SSIM | PSNR | LPIPS | Mem | SSIM | PSNR | LPIPS | Mem | SSIM | PSNR | LPIPS | Mem |
| $\tau_S$=0.01 | $\tau_p$=0.0008 | 0.817 | 27.38 | 0.212 | 374MB | 0.842 | 23.64 | 0.190 | 159MB | 0.902 | 29.55 | 0.249 | 265MB |
| $\tau_S$=0.001 | $\tau_p$=0.0008 | 0.815 | 27.41 | 0.211 | 450MB | 0.845 | 23.54 | 0.183 | 202MB | 0.903 | 29.69 | 0.241 | 380MB |
| $\tau_S$=0.01 | $\tau_p$=0.0004 | 0.820 | 27.52 | 0.202 | 637MB | 0.8531 | 23.86 | 0.173 | 314MB | 0.903 | 29.45 | 0.247 | 386MB |
| $\tau_S$=0.001 | $\tau_p$=0.0004 | 0.821 | 27.49 | 0.191 | 728MB | 0.853 | 23.73 | 0.162 | 304MB | 0.902 | 29.67 | 0.236 | 444MB |

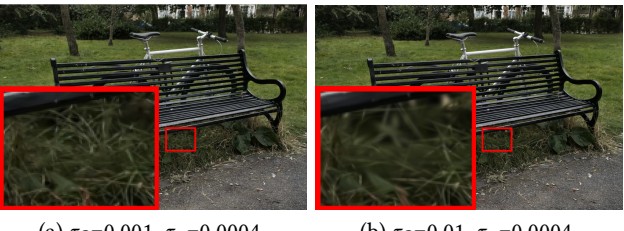

(a) $\tau_S$=0.001, $\tau_p$=0.0004      (b) $\tau_S$=0.01, $\tau_p$=0.0004

**Figure 8: Comparison of results for AbsGS under different scale thresholds.**

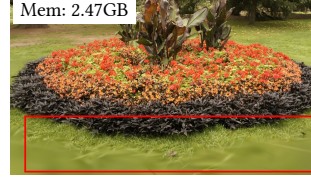

(a) Impact of $\tau_p$      (b) rendered image ($\tau_p$ = 0.0001)

**Figure 9: Comparison of results for 3D-GS under different gradient thresholds.**

at different thresholds $\tau_p$ in the *flowers* scene of Mip-NeRF360. The gradually increased memory requirements make the simple solution impractical for real-world applications. Additionally, we show rendering result when $\tau_p$ is set to 0.0001 in Fig. 9 (b). It's observed that the over-reconstruction issue is still evident even at the cost of 2.47GB memory. Further lowering the threshold below 0.0001 result in out of CUDA memory error when training a single V100 GPU. Above analysis reveals that simply lowering gradient threshold $\tau_p$ is impractical to eliminate over-reconstruction.

*Why does addressing over reconstruction require a large amount of memory for 3D-GS while AbsGS does not?* As illustrated above, directly reducing the threshold for 3D-GS $\tau_p$ is highly inefficient and impractical while our method manages to solve over-reconstruction. Next, we uncover the reasons behind it by visualizing selected Gaussians for split. In Fig. 6 (a), we train the *bicycle* scene with 3D-GS using default parameters for 3000 steps. The rendering image and ellipsoid image reveal that there are numerous over-reconstructed areas in the scene, such as lawn and trees. Fig. 6 (b) and (c) respectively show the Gaussians selected under different gradient thresholds for 3D-GS and AbsGS, with the number of selected Gaussians inset. In the case of 3D-GS with the default threshold of 0.0002, although it selected 77969 Gaussians, it did not effectively encompass over-reconstructed region. In contrast, AbsGS can select most of the over-reconstructed region by selecting only 37390 Gaussians at the threshold of 0.001. Besides this, lowering the threshold indeed allows for the selection of more over-reconstructed areas for 3D-GS, but the number of selected Gaussians increases rapidly, and the rate of noise in all selected Gaussians is also apparent, resulting in large memory cost. In particular, see Fig. 6 (b) and Fig. 6 (c), it can be observed that when we lower the threshold, 3D-GS selects many Gaussians that do not need split but still can not select large Gaussians in over-reconstructed regions. Above analysis demonstrates that the homodirectional gradient used by AbsGS offers a much

more accurate reflection of the representation quality, leading to fewer mistakenly selected Gaussians. This fundamental distinction proves why AbsGS is more efficient than 3D-GS.

## 5 CONCLUSION

3D-GS has made significant strides in novel view synthesis tasks, but it frequently encounters issues such as blurriness and loss of detail stemming from over-reconstruction. This paper delves into the phenomenon of over-reconstruction and identifies gradient collision in 3D-GS's adaptive density control strategy as a primary cause. Specifically, the view-space positional gradient used in this strategy is the sum of sub-gradients of all pixels covered by a Gaussian, and these gradient directions cannot always remain consistent, leading to mutual cancellation. To tackle this challenge, our proposed AbsGS utilizes homodirectional view-space positional gradient by taking the absolute values of the x and y components of pixel-wise sub-gradients separately, eliminating the influence of direction. Extensive experiments conducted on multiple datasets against 3D-GS demonstrate the significant advantages of our method in eliminating over-reconstruction and restoring fine details. Moreover, our approach boasts lower memory consumption compared to 3D-GS, largely attributable to AbsGS's more accurate identification of over-reconstruction through the use of homodirectional gradients.

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
