# OpenReview forum: "AbsGS: Recovering fine details in 3D Gaussian Splatting"
_acmmm.org/ACMMM/2024/Conference — MM2024 Poster_

### Official Review · Reviewer_PMWp · 2024-05-24

**Rating:** 5
**Confidence:** 3

**Summary:**

This paper proposes a novel densification strategy for 3D Gaussian splatting. Specifically, this paper proves that 3DGS fails to identify over-reconstructed regions because of the gradient collision incurred by its densification strategy. Then, this paper proposes to erase the effect of gradient direction to address this issue. Experiments results demonstrate that the proposed densification strategy can effectively remove artifacts caused by wrong identification of over-reconstructed regions.

**Strengths:**

1. The analysis of gradient collision is technically correct and easy to follow.
2. The proposed homodirectional gradient is simple yet effective.
3. The experimental results show that the proposed method can improve the effectiveness of densification, and thus, improve the rendering quality of over-reconstructed regions.
4. The paper is well-written and easy to follow.

**Limitations:**

1. It would be better to add the rendered image of the proposed method in Fig. 6 for comparison.
2. It would be better to discuss whether the same issues exist in the densification of under-constructed regions.
3. “as illustrated in 3.” in line 450, page 4 should be corrected to “as illustrated in Fig. 3”.

**Suitability:**

3

---

### Official Review · Reviewer_mMJA · 2024-05-24

**Rating:** 4
**Confidence:** 3

**Summary:**

This work proposes novel homodirectional gradient as the guidance for densification. The strategy efficiently identifies large Gaussians in over-reconstructed regions, and recovers fine details by splitting.

**Strengths:**

The work was novel and the experiment valid. The visual proof is sufficient to show that the method is effective for the problem.

**Limitations:**

1.the relatively low improvement of this method is not enough to affect the contribution of this paper, and it is suggested that the motivation of gradient conflict should be further elaborated.
2. It is highly recommended to validate this generic approach against other Gs-based methods, such as Scaffold GS

**Suitability:**

3

---

### Official Review · Reviewer_4HYB · 2024-05-24

**Rating:** 5
**Confidence:** 4

**Summary:**

This paper introduces a novel densification scheme for 3D Gaussian Splatting, utilizing homodirectional gradients (taking absolute value) as guidance. The proposed method enhances rendering quality and achieves a more reasonable distribution of Gaussians, particularly in regions rich in details.

**Strengths:**

1. This paper focuses on an important issue of blurriness in certain areas of 3D Gaussian Splatting (3DGS), which I also noticed when I first used 3DGS.
2. The solution presented in this paper is simple and effective (I happened to try it before reviewing the paper). I appreciate that the code of AbsGS is open source.

**Limitations:**

Some confusing points:
1) Why do the authors describe the phenomenon where "regions containing high-frequency details are covered by only a small number of large Gaussians" as "suffering from over-reconstruction"? Typically, over-reconstruction implies the use of too many primitives.
2) In Line 454, the authors mention that the "over-reconstruction" phenomenon is shown in Fig. 3(c). However, it is unclear how Fig. 3(c) was derived.
3) The number of Gaussians displayed in Fig. 1 is not consistent with the memory usage reported in Table 1. The proposed method in Fig.1 uses more gaussians while the average memory of each dataset in Table 1 is less than 3DGS. I understand that the results shown in Fig. 1 are calculated for a single scene, but I am curious whether this discrepancy is due to random variation or if there are other underlying reasons.
4) Fig.9 (a) shows the change of LPIPS as the threshold changes. What about the PSNR and SSIM?

Some typos, such as:
1) "Gaussion" in the caption of Fig.1.

**Suitability:**

3

---

### Official Review · Reviewer_sF6u · 2024-05-26

**Rating:** 5
**Confidence:** 4

**Summary:**

The paper propose an important densifycation strategy for 3DGS(3D Gaussian Splatting) to recover the details of the scene. The paper provide detail analysis of the comparision between the previous densification strategy and proposed densification strategy. The experiments demonstrated that the efficiency of the proposed method.

**Strengths:**

* The paper explored an important problem and propose a novel idea to solve the problem. The details of the reconstructed scene can be successfully recoverd using the proposed densification strategy.

* The paper writing is clear and easy to follow.

**Limitations:**

There are some points in the paper are confusing and should be solved from the authors feedbacks.

* I wonder whether the proposed methods can not only limit to the 3D ellipsoid representation(3DGS)? Will the proposed densification strategy still be useful for 2D-surfel representation (eg. 2D Gaussian Splatting for Geometrically Accurate Radiance Fields, Huang et. al
)? I suggest the authors to provide some experiments and discuss about that. Since 2DGS is a new representation that have the trend to replace the 3DGS. Otherwise, the paper looks really like an incremental on 3DGS. If the proposed methods can still be used for different representation. I believe it will make the paper more fundamental.

* The memory usage of AbsGS. From the Figure 4(d,e). I notice that the points marked as densification points of the AbsGS is much less than original 3DGS(69269 vs 16448). However, I do not see significant decrease of the memory usage of the proposed methods (AbsGS-004 vs 3D-GS in table 1). I need some clarifications about that.

* Why the memory usage of AbsGS performs much different across 3 different dataset.  I notice that in DeepBlending dataset, the proposed methods decrease the memory usage most significantly while other two datasets do not reveal this. I wonder what is the reason about that.

* The paper miss some experiments on the NeRF-synthetic dataset,  does this mean that on synthetic dataset, absGS do not have significant improvement compared with 3DGS. I understand that the object in NeRF-synthetic dataset is much easier to reconstruct.

**Suitability:**

3

---

### Meta-Review · Area_Chair_bU1d · 2024-07-08

**Recommendation:** Accept (Poster)
**Confidence:** 5

**Metareview:**

This paper presents a comprehensive analysis of the cause of aforementioned artifacts called as gradient collision, which prevent large Gaussians that cover small-scale geometry from splitting. The proposed method is effective and new. All reviewers have positive final ratings, are satisfied with the response, and recommend accepting the paper. I agree with their recommendation. Thanks for the authors' effort and rebuttal.